# Supporting Explanations Within an Instruction Giving Framework

**Alan Lindsay** and **Ronald P. A. Petrick**

Automated Planning Lab,
Department of Computer Science,
Heriot-Watt University, Scotland, UK
{alan.lindsay,r.petrick}@hw.ac.uk

## Abstract

As AI Planning has matured and become more applicable to real world scenarios, there has been an increased focus in explainable planning (XAIP) (Fox, Long, and Magazzeni 2017), which focuses on making the planning model, process and resulting plan more explainable. In the context of a plan-based instruction giving agent, explainable planning is a vital ingredient in supporting agents to be capable of effective interaction, as explaining aspects relating to the plan, or model form natural parts of an interaction. As a starting point we have considered the analysis of a corpus of task based human human interactions. This analysis identifies transactions (roughly plan steps) as key components within the interaction, where parts of the interaction will largely focus on the specific step (e.g., instruction) under consideration. We have developed a new framework that exploits this structure, by organising the interactions into a series of loosely coupled transactions. In this framework explanations play an important part both at the transaction level (e.g., instruction clarifications) and at the task level (e.g., intention). We have developed a prototype system, which can support large scale interactions. Our results also indicate that our system can be used to elicit information from the user at execution time and use this information to select an appropriate plan. We show that this can lead to fewer explanations.

## Introduction

Automated planning has been identified as an appropriate technology for creating both task-oriented dialogues (Muise et al. 2019) and task-based interactions (Petrick and Foster 2013). In these domains an agent may guide the user through a plan, describing each step in the plan, answering queries and perhaps justifying its decisions or intimating its intentions or expectations. Using automated planning to underpin agent human interaction can provide a more compact means of capturing interaction structures and the model based approach also supports verification of the possible interactions, which is not currently provided by data driven approaches (Muise et al. 2019). A current limitation in plan based approaches is that they typically synthesise a plan for the entire set of possible interactions, which can limit the size of feasible interactions.

In our work we have observed that in task based interactions, such as a Tour Guide scenario (Petrick, Dalzel-Job, and Hill 2019), where an agent guides a user through a city visiting various landmarks, there will be many similar interaction episodes. For example, instructions about how to move between two points will often share similar structure and user queries will typically be similar (e.g., 'how far?', or 'is it past the cafe?'). Our framework is based on chaining together small transactions, which are focused on a single step in the task (e.g., instructing the user and managing the associated questions and uncertainties). Underpinning the chaining of these transactions is a new planning approach (Lindsay, Craenen, and Petrick 2021), which supports a type of within task elicitation. This approach allows information elicited from the user (e.g., preferences or knowledge) during execution of the plan to directly impact on the selection of an appropriate action sequence for the user.

Explanations form a natural part of task based interactions. For example, in a corpus of interactions captured for a human human instruction giving task, various types of explanations are typically used during the interactions, including clarifications, answering direct queries and providing explanations. This suggests that explanations may form an important part of interaction agents and corpora of existing task based interactions can provide guidance in the design of these agents. We examine how the types of explanations that are demonstrated in task based interactions can be supported in our framework. In particular, we exploit the natural abstraction supported by the two layer architecture. Thus considering the specific instruction based queries within the small transactions and using the task plan to support more general queries, e.g., regarding the agent's intention.

In this paper we examine an analysis of a corpus of task based interaction dialogues and consider the types of explanations that were used during the interactions. This analysis is used as the motivation for the construction of a transaction domain model, for managing a focused interaction around a single plan step. Each of these transactions exists within a complete task and we consider how task level explanations can also be supported. In particular, we consider that the hierarchy provides a natural division that can be exploited in the generation of more concise explanations. We have implemented a prototype system and used it to simulate interactions in a Tour Guide setting. The simulations demonstrate

that our system is capable of selecting appropriate action sequences for different user types, is able to support long interactions ($> 150$ steps) and can support explanation generation and query answering.

The paper is structured as follows: we first present the related work and an example domain; we then overview an analysis of a relevant corpus, we present the transaction domain model and our framework for supporting interactions; we then present our prototype system and observations; we finish with our conclusions.

## Related Work

Our aim is to develop a general interaction agent that performs socially appropriate behaviour during interactions between a robot and a human, extending the approach developed in (Petrick and Foster 2013). A key component of this is in ensuring that the agent is able to explain its decisions (Langley et al. 2017). Inspired by (Madumal et al. 2019) we start from example interaction dialogues. However, whereas (Madumal et al. 2019) focus on explanation dialogue, we use full task based interactions. Our current approach is based on capturing the requirement of explanation generation as part of the modelling process —based on the structures extracted from the interaction dialogues— as opposed to identifying them based on e.g., a model of the user's understanding of the world (Chakraborti et al. 2017).

Frameworks that support human agent interactions have used various planning technologies including epistemic (Petrick and Foster 2013), FOND (Muise et al. 2019) and Hierarchical Task Network (Behnke et al. 2020) planning. In (Behnke et al. 2020) the user can add new constraints into the planning model during execution, resulting in replanning. This user input is handled by the framework and is not reasoned about by the planner. In (Petrick and Foster 2013; Muise et al. 2019) contingent plans are constructed to support within task elicitation, allowing the execution to be influenced by the user. A key difference in our approach is that the elicited information is used to impact the utility function and not the causal structure of the problem. This provides a rich language for capturing important trade-offs, between e.g., action costs, user preferences and the costs of explanation and elicitation (e.g., annoyance). While our approach to planning with preferences (Jorge, McIlraith, and others 2008) focuses on eliciting preferences within the task execution, other work has supported user influence at plan time (Das et al. 2018), or combining elicitation and planning/execution within a single framework, such as the factory setting, user tailoring, execution tuning (FUTE) framework (Canal, Alenyà, and Torras 2016)

### Instruction Giving Scenarios

We consider an instruction giving and following scenario, where an instruction giver guides an instruction follower through a series of steps and manages the associated responses, e.g., uncertainty misunderstanding, or disagreement. In this work, we underpin the agent's decisions and behaviour with a planning based approach that is used both to generate the agent's strategy, as well as construct situation-based instructions and explanations. We now

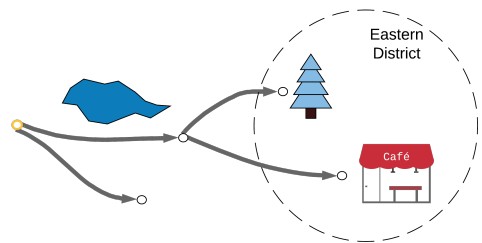

Figure 1: An example instruction giving scenario, with landmarks that could be used to guide the instruction follower.

| G | "you're going to do a backward *S*.." | [Instruct] |
|---|---|---|
| F | "right" | [Ack] |
| G | "right" | [Ready] |
| G | "you're going to go down slightly for about a centimetre" | [Instruct] |
| F | "is this before or after the backward *S*?" | [Query-w] |
| G | "this is before it" | [Reply-w] |
| F | "right" | [Ack] |

Table 1: Part of a dialogue from the HCRC Map Task (Anderson et al. 1991) corpus.

present an instruction giving scenario that will be used for the examples in this work.

**Tour Guide Domain** A tour guide agent directs a user through a tour of e.g., a town, stopping at the important landmarks on the route. Designing the tourist experience has been compared to storytelling in (Stienmetz et al. 2020), with the plot as a chaining together of interactions or events, which must balance various factors. As with storytelling, different user's will appreciate the various possible landmarks differently. For instance, consider a tourist on a guided walking tour of a city (example from (Petrick, Dalzel-Job, and Hill 2019)). After reaching a place where they can see they are almost back to the starting point, the tour guide says "Let's go up that hill," pointing to a large hill. "We can get a good view of the city from there." However, on seeing the tired expression on the tourist's face, the guide adds "Or we can stop at that cafe over there and take a break." This is an example where it would be necessary for the agent to be able to use information elicited at execution time in order to influence the remainder of the execution.

## Explanations in Interactions

In this section we consider the role of explanations within task-based interactions. We use an existing corpus of task-based interactions to support our study and first introduce the task and the corpus. We then examine the types of explanations that occur within these interactions, providing examples from the corpus.

### Interaction Transactions

We focus on the HCRC Map Task corpus (Anderson et al. 1991), which is a collection of dialogues from pairs of hu-

man participants performing the Map Task. In the Map Task, two participants are each given a map and a role: either the instruction giver, or instruction follower. Each map presents a series of landmarks and the instruction giver's map includes a route through the map. The task involves the instruction giver providing the instruction follower with instructions so that they can follow the route on their map. The maps can vary from each other, with missing or misplaced landmarks. The interactions for the Map Task exhibit a variety of conversational components, including instructions, queries, responses, explanations and alignment.

The corpus has been analysed using a coding system (Carletta et al. 1996), in order to examine the dynamics of dialogue. This analysis identifies move, game and transaction structures, which form a hierarchy, and organise the dialogue into task orientated components. The atomic blocks of the coding system are the conversational *moves*. The coding system identifies twelve moves, including the *instruct* move and the *acknowledge* move. Table 1 presents part of a dialogue from the corpus (Anderson et al. 1991) and indicates the speaker (G: instruction giver, F: instruction follower), and the move, e.g., *instruct* or *acknowledge* (Ack). The first utterance is from the instruction giver (G) and provides an instruction for the follower. There are query moves, such as *Query-yn* and *Query-w* and corresponding reply moves (e.g., *Reply-y*, *Reply-n* and *Reply-w*). Any question to the partner that is answered yes or no, is a *Query-yn* move. Any other question is a *Query-w* move (e.g., see the follower's second entry). This dialogue is an example of a query used to determine whether the instruction follower had a specific landmark on their map.

| G | "Do you have the west lake, down to your left? | [Query-yn] |
| F | "No." | [Reply-n] |

These conversational moves are organised into conversational games. Each game groups together the moves that start at some initiation move, such as an instruction, query or explanation, and end with a resolution, e.g., an acknowledgement, or failure. At its simplest, a game will constitute a question and its answer, or an instruction and acknowledgement. However, the games will often be more involved and can be embedded.

The transaction coding is based on identifying how the instruction giver has broken the route into pieces, as part of their strategy for communicating the route. This process divides the task into a sequence of major steps in the instruction giver's route. A transaction gathers the conversational games that pertain to one of these steps. In the example in Table 1, the instruction giver provides the follower with an overview of the next section of the route and then begins to explain the shape in more detail.

### Explanations in Transactions

Explanations form an important part of transactions. The instruction giver can provide the follower with additional information, or clarification. And the follower can explain the implementation of the instruction: describing their approach and the resulting observations. We will now examine some

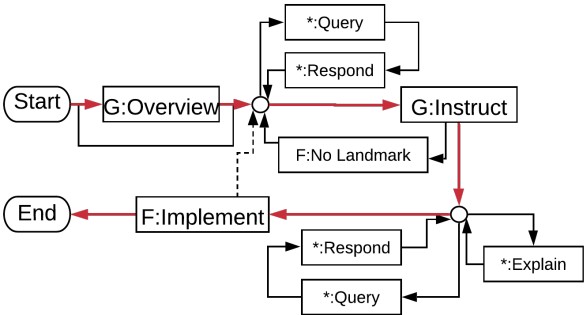

Figure 2: An abstraction of the transaction domain model structure. The red edges indicate the steps in a straightforward, single step move or visit instruction transaction.

of these types of explanation with examples from the Map Task corpus.

The specific *explain* move in the conversational move coding system, involves providing additional information that was not specifically elicited. For example, in this dialogue the instruction follower accepts an instruction then provides an explanation of what they are doing.

| G | "and then down 'til you're right below the bottom of the dead tree | [Instruct] |
| F | "right" | [Ack] |
| F | "I have to jump a stream." | [Explain] |

Explanations can also provide clarification of an instruction given, where the listener cannot understand what was intended. For example, in Table 1 the instruction giver presents an instruction to the follower. The follower is unclear about the intended sequence and asks for clarification. Similarly, in the following example, the follower attempts to clarify an instruction from the giver. However, the follower uses a landmark that is not on the instruction giver's map. This is explained to the follower in the giver's reply.

| F | "Is down three steps below or above the machete?" | [Query-w] |
| G | "The machete's not on my map." | [Reply-w] |
| F | "Oh" | [Ack] |

Another common explanation is where the instruction giver provides an overview of the following section of the route (e.g., see the first line of Table 1). In some examples, the instruction giver provides a longer overview, which incorporates several sections of the route. Although in the Map Task, the route is provided to the instruction giver, similar explanations have been used to communicate an agent's intentions (Foster et al. 2009; Lindsay et al. 2020a).

### Instruction Giving Transaction Model

In this section we present our model for capturing the interaction for a specific transaction. As was mentioned in the previous section, a transaction captures a sub-dialogue associated with a specific step in the task. Therefore there is a very specific context for each transaction, such as 'move from the lake to the forest', or 'visit the museum'.

```
(:action instruct_follower_using_landmark
  :parameters (?source-pos - position ?step - step
                 ?l - landmark)
  :preconditions (and (follower-located)
                 (source ?source-pos ?step)
                 (relies-on ?l ?step)
                 (follower-at ?source-pos)
                 (not (nk-f-landmark ?l))
                 (not (open-instruction)))
  :effects (and (have-instructed-follower ?step)
             (open-instruction)))

(action query_landmark_exists_yn
  :parameters (?source-pos - position ?step - step
                 ?l - landmark)
  :preconditions (and ...)
  :effects (and (have-questioned-follower ?l)))

(action reply_landmark_query_n
  :parameters (?source-pos - position ?step - step
                 ?l - landmark)
  :preconditions (and ...
                 (not (nk-f-landmark ?l))
                 (have-questioned-follower ?l))
  :effects (and (nk-f-landmark ?l)
             (resolved-query ?l)
             (not (have-questioned-follower ?l))))
```

Figure 3: Examples of instruct, query and response PDDL actions. The alternative instantiations of the actions is determined by the specific scenario and represented in the problem model. Similar preconditions have been contracted.

We have used the move and game structures from the dialogues in the HCRC Map Task corpus (Anderson et al. 1991) to support the design of a planning domain model. The domain model captures the interaction as a series of potential action sequences (see Figure 2). For example, it includes actions for instruction giving (*Instruct*), querying and responding (*Query* and *Respond*) and an action *Implement*, which represents the actual implementation of the instruction. Each of these graph nodes represents several alternative moves in the dialogue structure. For example, the *Instruct* node is implemented using several operators, including: *instruct_follower_using_landmark*, an instruction that relies on a specific landmark; *instruct_follower_using_directions*, which is an instruction using directions and distances instead; and *instruct_follower_visit*, which instructs the user to visit a POI (point of interest) at their current position. Figure 3 (top) presents the PDDL action representing an instruction that relies on a landmark (the parameter, *?l*). The parameters include the follower's current position (*?source-pos*) and a step object (*?step*), which represents an available instruction, consisting of a source and destination position. These alternative instructions are represented in the problem model as alternative steps and each step is associated with its required landmarks (as appropriate). More generally the problem model captures the appropriate alternative instructions, queries, responses, explanations, and overviews for the specific transaction scenario. For example, for a move

| G | "We're heading East towards a Cafe." | [G:Overview] |
| G | "Go past the lake and stop at the fork in the road." | [G:Instruct] |
| F | "I can't see a lake." | [F:No Land'] |
| G | "Oh, ok." | [G:Ack'] |
| G | "Go East for a few hundred meters and stop at the fork in the road." | [G:Instruct] |
| F | "Ok." ⟨Moves⟩ | [F:Implement'] |
| G | "Go East towards the Cafe." | [G:Instruct] |
| | ... | |

Table 2: Example of an interaction captured by the transaction planning model, for the scenario in Figure 1. Labels refer to nodes in Figure 2.

instruction, the instruction giver can query whether the follower knows about any of the landmarks that is required to make progress from the current position. In Figure 2, these queries are represented by the query *G:Query* before the instruction is given. Once the instruction giver is aware that the follower does not have a landmark: *(nk-f-landmark ?l)*, then the associated instructions cannot be used. This can be found out through a direct query (see the query and response actions in Figure 3), or in response to the instruction giver's instruction (see the *No Landmark* node in Figure 2).

The model captures the possible interactions for each transaction. Of course, the interaction relies on both the instruction giver and follower. We have considered a simple executive that chooses an intended plan and follows it, implementing the intended actions (those prefixed with 'G'). At points in the model with possible follower input, the executive accepts input and transitions following the follower's input. This approach has proven sufficient to support a diverse range of alternative interactions (we consider this further in the discussion).

**Transaction Specific Explanations**

At the start of each transaction a problem model is defined, describing the available instructions (microsteps), queries, responses and explanations. For example, the interaction presented in Table 2, both the initial instruction ("Go past the lake..") and the alternative instruction, which relies on no landmarks ("Go East..."), are each encoded as possible steps in the planning model. In this work, we have described these by hand for each of the transactions. However, we aim to explore the potential to populate these explanations (semi-) automatically. For example, utilising plan and model based explanations. For example, the *G:Overview* node could be implemented using approaches that provide an explanation of the agent's intent (Foster et al. 2009; Lindsay et al. 2020a).

## A framework for Supporting Interactions in Planning

A characterising property of the instruction giving domains is the interleaving of dialogue and task aspects during the interaction. In the Map Task structure coding, transaction structures organise the dialogue around specific steps in the

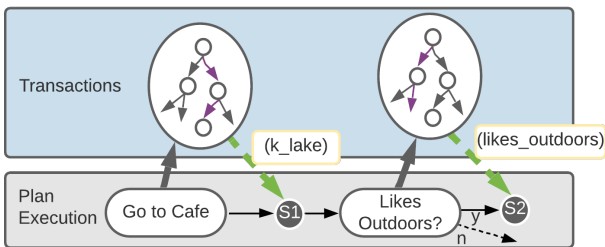

Figure 4: The framework supports loose coupling of transactions, which mostly focus on the specific instruction. Only specific information is retained from the interaction. E.g., an interaction for moving to the cafe in Figure 1 might discover that the user knows about the lake.

task. Each transaction can therefore be viewed as corresponding to the dialogue required for a single plan step. Each of these transactions forms a largely independent component. To this end we have designed a layered approach for managing the agent's task strategy and their transactions with the user. We have used the structures created for the Map Task to inform the design of dialogue structures for the related Tour Guide domain. We have adopted a new within task elicitation planning strategy (Lindsay, Craenen, and Petrick 2021), which allows dependency between the transactions and the selected plan. In this section we present our agent's framework based on the dialogue transactions.

### Chaining User Agent Transactions

Tasks like the HCRC Map Task (Anderson et al. 1991), the robot bartender domain (Petrick and Foster 2013) and the tour guide domain (Petrick, Dalzel-Job, and Hill 2019), involve sequences of transactions, where the agent must make progress towards an overall goal, while at the same time managing local uncertainty and conflict. We observe that in domains such as the tour guide domain, much of the interaction required to manage the user's understanding of a particular stage will be irrelevant to the remainder of the task. Within each transaction of the human-human transcripts for the HCRC Map Task, the dialogue largely focused on the relevant part of the plan (Carletta et al. 1996). As we discussed in the previous section, this involved providing instructions, responding to potential queries and using alternative strategies for describing the instructions, as necessary.

Consequently we have adopted a layered framework, where the dialogue transactions are abstracted from the main task structure. Our framework has two main components: the agent's task view and a set of transaction models (see Figure 4). The agent's task view is captured in a planning model, which provides its model of the world. This model provides the goals to be achieved and is used by the agent to generate a plan (its intended course of action). Each dialogue action in the domain is associated with a *transaction model* (see the previous section), which captures the possible agent user interactions for the particular type of action. During execution the agent's plan steps are implemented by invoking the appropriate transaction model, which manages the implementation of the plan step.

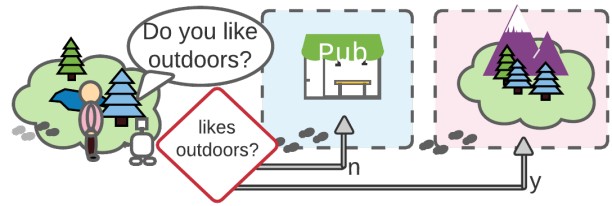

Figure 5: During the task the robot is able to elicit whether the user likes outdoors or not. This is then used in subsequent decisions of what activities should be planned.

### Within Task Elicitation

In order that an agent can be an effective collaborator it is important that the agent is able to adapt its behaviour for the knowledge and preferences of a particular user. Consider again the guided walking tour example, where the tour guide is able to query the user at a key point in the tour in order that the tour can be better suited to the user. We observe that in many situations it will not be appropriate for an isolated elicitation process to proceed the execution of the task, as is commonly assumed for user preferences, e.g., (Boutilier 2002; Boutilier et al. 2006).

Our approach is to model a set of user observations as hidden state variables that can be elicited by the agent. For instance, the instruction giver might discover information about the landmarks that the follower knows about, or can see. This is demonstrated in the Map Task interactions, where the maps share some landmarks. Retaining the instruction follower's knowledge of landmarks between transactions would allow the instruction giver to consider how to best instruct the follower in subsequent transactions. In the tour guide scenario, the important user observations might also include the user's preferences for certain types of landmark, e.g., outdoors, social or educational. These preferences and knowledge can be modelled by a set of Boolean variables, e.g., {*likes-educational$^U$*,*likes-social$^U$*,*k-blue-tree$^U$*,..}, which are unknown in the initial state. At the most basic a transaction will be associated with a sensing action and the instruction giver will directly elicit the value from the follower. However, as we saw in the previous section, some transactions will include both knowledge and ontic effects.

Our framework extends net benefit planning (Menkes Van Den Briel, Do, and Kambhampati 2004) with partial observability (Bonet and Geffner 2011). The aim in net benefit planning is to identify a sequence of actions that maximises the net benefit, which is calculated by removing the action cost from the plan's utility. A key idea to enable within task elicitation is to make the utility function for the net benefit problem dependent on the user's attributes. For example, in the tour guide scenario, the utility of selecting to visit a pub as a landmark on a tour will depend on whether the user likes socialising (represented by *likes-social$^U$*):

$$u(pub^A) = \begin{cases} 40 & pub^A = \texttt{true} \bigwedge likes\text{-}social^U = \texttt{true} \\ 20 & pub^A = \texttt{true} \bigwedge likes\text{-}social^U = \texttt{false} \\ 0 & otherwise \end{cases}$$

Thus the planner selects which information should be elicited and when to elicit it, in order to distinguish between user types. This allows the planner to customise the appropriate parts of the contingency tree and optimise the action sequence for those users.

### Global Task Explanations

In the Map Task, the instruction giver was presented with a route to describe. There are examples where the follower does question the route. However, as demonstrated in the following example, the instruction giver is not really able to provide much justification.

| G | "...before you come to the bakery do another wee lump" | [Instruct] |
| F | "Why?" | [Query-w] |
| G | "Because I say." | [Reply-w] |

The situation is different in our framework, as it actively selects the plan to follow. We are therefore able to exploit XAIP techniques in order to extend the scope of the explanations, presented in the previous section, to include global plan level explanations. In this section we consider the available context and the generation of appropriate explanations.

At the point of populating the next transaction model, the system has a rich context for generating task level explanations. This includes the current state of the world, the known and unknown user attributes and the agent's planning model and the agent's intention for how the interaction will progress. This context is sufficient for several current approaches to XAIP, e.g., (Krarup et al. 2019; Eifler et al. 2020; Lindsay et al. 2020a).

The agent's intentions are captured in its plan, and as such providing a view of the plan to the instruction follower allows them to better understand the agent's intentions. In the Map Task interactions, some instruction givers provided the follower with an overview of several plan steps in advance to prepare them for the upcoming instructions. In our current system we adopt this strategy to communicate the agent's current intentions, in order to provide motivation for a particular instruction. Of course, communicating the agent's complete plan will often not be appropriate as part of an interaction. For the purposes of our user study we used the next target, or subgoal in the plan to provide a localised view of the plan (Lindsay et al. 2020a). For example, in Figure 1 the agent might have a plan to pass the lake and then move on to visit the cafe. In this example, the identified target is the cafe, which is the next landmark to visit (e.g., 'We're heading to the Eastern District to visit a cafe.'). As well as providing justification it also provides more context for the user, allowing them to identify existing conflicts with how they see the task. This may also provide a starting point for an exploration of alternatives, e.g., through contrastive explanations (Krarup et al. 2019).

## System Demonstration

In this section we present our prototype system and provide initial preliminary results. We have used the Tour Guide domain as a case study and designed an interaction for this domain. We first describe the system implementation, we then

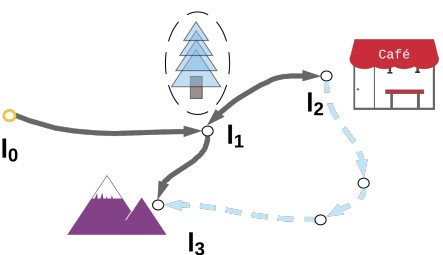

Figure 6: The tree (landmark) at $l_1$ might not be visible. If it is not then navigating between $l_2$ and $l_3$ using a longer path (the blue dotted line) might require fewer explanations.

look at the transactions that are supported and use the system to simulate complete interactions. We conclude the section with a discussion of the approach.

### The Prototype System

We have implemented a prototype system that can manage interaction action sequences that chain together transactions. Our framework incorporates two main components: an implementation of within task elicitation (Lindsay, Craenen, and Petrick 2021), which extends the K-Replanner system (Bonet and Geffner 2011); and a system that manages the communication between the task level and the transaction problem models, which wraps the LAMA-11 configuration of Fast Downwards (Richter and Westphal 2010). The K-Replanner system was extended to support cost sensitive planning and to generate the complete contingency tree. We compare this approach to a Baseline, which uses LAMA-11 directly and cannot use elicitation. The system provides modelling support for specifying a utility function that depends on the user attributes.

The output of the modified K-Replanner system is a complete contingency tree, with actions and sensing actions. This tree is read in and implemented by an executive, which incrementally steps through the actions and chooses the correct branches in turn. Each action is implemented by selecting and completing the appropriate transaction problem model for the current situation and the action type. The problem models for each of the possible transactions are largely specified upfront as part of the construction of the scenario. For example, the available instructions and their dependencies, the types of queries and responses, the initial position and the goal are all specified for each of the move or visit actions in the Tour Guide domain, as well as the sensing actions. The problem is finalised by including the current user attribute values, which allows continuity between transactions. For example, in the Map Task, the user attributes note whether it is known that a certain landmark is known to the instruction follower. Our executive implementation has both an automatic mode and an interactive mode, which allows the follower's actions to be selected during each interaction.

After each transaction the final state is analysed to determine if values have been found for any user attributes. These values are noted by the executive and the executive also implements the applied action, updating the task level state.

The user attributes discovered during these transactions are used to select the appropriate sensing action branch in the contingency tree. If the executive discovers a sensing action that has no associated value, the transaction associated with the sensing action is selected. The problem models for sensing the e.g., instruction follower's preferences in the Tour Guide domain involve a goal of having discovered the value of the specific user attribute. These queries are implemented using a subsection of the transaction domain model and can involve queries, responses and explanations.

This approach opens the possibility that the agent knows the value of an attribute before its value is used to branch in the plan. In situations where the elicitation is guaranteed during the transaction associated with an action then the sensing action can be conditioned to be triggered by the application of the action. It is therefore in situations where the elicitation is uncertain. It will be interesting to consider the possibility of setting more specific goals within the transaction models, and allowing the task planner more choice, by using joint ontic and sensing actions at the task level.

## Preliminary Investigation

As a case study we have used a Tour Guide scenario, where we simulate an agent instruction giver providing instructions to a human instruction follower. We assume that the instruction follower will have preferences for certain landmarks, which are captured in three user attributes: {*likes-outdoors$^U$*,*likes-educational$^U$*,*likes-social$^U$*}. We also include the possibility that the follower does not have knowledge of certain landmarks (either not on the map, or are not visible) and these are captured in an additional two attributes: {*k-blue-tree$^U$*,*k-lake$^U$*}. In this scenario, the instruction giver will provide instructions for the follower to move between points and visit certain landmarks. Each of the movements are associated with a transaction model and can be divided into one or more steps (instructions).

**The Agents:**  To test the system we created two agents: an instruction giver and follower. The instruction giver is a plan based agent, which uses replanning when its current plan is not applicable. The behaviour of the instruction follower is parameterised to control its behaviour during an interaction. These parameters include the probability that an explanation is given or an agent queries something (the instruction, or the situation). Separate probabilities are used for landmark and direction based instructions. There is also a probability that the follower questions the chosen plan, which depends on the utility of the next subgoal. The probabilities have been estimated using example Map Task dialogues.

**Task Level**  We examine three task level scenarios: two small examples to demonstrate specific features and a third to examine how the approach applies to large problems.

**1. Responding to Within Task Elicitation**  To demonstrate the within task implementation we implemented the scenario with a lake, a forest and a pub, presented in Figure 1. The forest and pub are provided as possible places to visit and the directed graph prevents both being visited. A plan was generated that included branching on likes/dislikes for the landmark types. The plan first branches on whether the user likes outdoors activities. If they do then the utility of visiting the forest is higher in the utility model (in our example, it has utility 7 over a maximum of 5 for the pub). Otherwise, the plan indicates moving to the pub. At the pub the plan branches on whether the user likes social activities. If they do then the pub is visited (5 utility). Otherwise the pub is skipped (as the net-benefit of visiting is not above zero). This contingent plan therefore captures suitable action sequences for each of the user types.

**2. Improving Explicability**  We have created a scenario inspired by the Map Task, where a landmark ($l_1$) can potentially be missing from the user's map (see Figure 6). In the case that the landmark is missing then transactions that pass the landmark are more likely to be longer and require more queries and explanations. Consequently, if the user does not know about $l_1$ then there is a cost associated with instructing the user to $l_1$. In the task the shortest plan involves passing $l_1$ several times. However, there is a longer path from $l_2$ to $l_3$ that is not reliant on $l_1$ (i.e., it is straightforward to explain). We used the within task elicitation approach to generate a branched plan for the scenario. The first action in the plan moves from $l_0$ to $l_1$ and discovers whether the user knows of $l_1$ or not. In each case the plan branches visit $l_2$ next. If the user knows $l_1$ then the plan returns to $l_1$ on route to visit $l_3$. Otherwise, the plan uses the longer route (due to lower cost for user's that do not know $l_1$). This example demonstrates how the information gained in transactions is used to influence the chosen path; using the fact that if the user is missing the landmark then the transactions are more likely to result in confusion and more queries.

**3. Task Based Interactions**  We have created a larger Tour Guide scenario, using a map from a Map Task interaction as inspiration. The map has 13 *visitable* landmarks defined in the task level problem, separated into outdoor, educational and social types. Overall, there are 51 positions, including the possible steps involved in multi-instruction transactions. The utility and cost models were designed so that users that liked a certain type would associate it with higher utility. A contingent plan was generated for this problem, with branching points for testing user attributes. The contingent plan has 154 nodes and took around two hours to construct. The average number of plan steps (including branching points) is 28.25 for each type of user (baseline has 25 steps). Using a hand-crafted cost model, there was a 6.18% reduction when using our within task elicitation method, compared with the baseline strategy. This is based on summing the cost for each type of user for the plans generated by each plan. This demonstrates that the within task elicitation is allowing important information to be elicited and exploited within the task level planning model.

## Transaction Level

Starting with the plan generated for the largest of the problems (#3), we generated 100 interaction simulations. A

random type of user was first picked and then interaction was simulated using a parameterised follower agent (see the agents descriptions above). The average total number of steps (including transaction micro steps) was 157.66 (SD=63.44) steps and the average simulation time was 11.94s (SD=5.53). This included an average of 44.45 (SD=20.49) plans or replans to simulate the transactions. These are therefore relatively large interactions (e.g., comparing favourably with the examples of interaction trees in (Muise et al. 2019)).

**Generated Explanations**    In this part we assume a mapping process taking the instantiated actions to natural language (NL) sentences. For the purposes of this work, this process was carried out by hand. As an example, we generated NL mappings for the small problem presented in Figure 1, and simulated transitions for this problem. One of the generated interactions was presented as an example in Table 2.

Of the 100 simulations generated for the larger task, 686 distinct transactions were generated. In a similar way to the example in Table 2, these included queries, explanations, and elicitations.

**Overview Explanations**    The overview explanations are based on the sequence of task plan steps to the next target or subgoal (Lindsay et al. 2020a) and each plan node is associated with an explanation. The target is identified by looking forward in the tree until either a goal is achieved, or knowledge is gained. For example, in Figure 1 the agent's intention is communicated as an indication of the next intended place that will be visited, which is the cafe (the explanation can be further refined to indicate the region etc.), e.g., "We're heading to visit the cafe, it's the next landmark in my plan." In certain settings it might be appropriate to overview the upcoming plan steps, e.g., ".. First we'll pass a lake and then..." In the Map Task, interaction overviews typically outline several steps and these steps could be identified from the plan.

### Discussion and Future Work

The transaction models described in this work have the flexibility to support various useful types of transaction, as well as some of the ideas that we intend to incorporate in the future. However, they have limitations and so, we consider alternatives that would allow different types of interaction and might be fitted into our modular framework. The use of a contingent planner (Petrick and Bacchus 2002; Muise et al. 2019) within each transaction would allow the planner to reason directly with the potential user responses. The use of epistemic planning models would provide more natural modelling of the knowledge of the instruction follower, as well as powerful inferences, which might lead to simplified interactions, e.g., (Petrick and Foster 2013). We are also interested in approaches for populating the transaction model automatically, exploiting clues in the domain model structure (Lindsay 2019) or additional knowledge structures (Vallati, McCluskey, and Chrpa 2018).

We are currently conducting a user study to investigate user response during an interaction with an agent (Lindsay et al. 2020b; 2020a). Part of this study is investigating predictors of human confusion when interacting with the virtual agent. Within a transaction, we are interested in using these predictors to act as triggers for certain action types. For example, on observing that the follower appears confused on hearing an instruction, the instruction giver might elaborate on the instruction, or use one of the alternative methods of explaining the microstep.

In this work we assume that the given instructions are followed by the instruction follower. Within each transaction the system uses a replanning approach and can support the user making alternative micro moves. However, it is assumed that the user will eventually satisfy the original plan step. The follower only influences the plan in situations where the instruction giver elicits their preferences. One approach is to extend the contingency tree with $k$ user *errors* (Lindsay et al. 2020b), allowing more of the contingencies to be expanded offline. The within task elicitation planner is based on K-Replanner, which is originally an online planner, so the approach could be easily adapted to allow replanning during execution.

Modelling these problems, involves creating appropriate task and transaction level problem models. This is assisted by the modular approach, which provides structure for creating interactions based on the repetitive tasks. However, appropriate utility and cost models must be created for each problem. We have considered this problem at the task level (Lindsay, Craenen, and Petrick 2021) , but are interested in investigating this for all models, in the context of existing approaches to domain model acquisition for natural language (Lindsay et al. 2017), interactions (Sreedharan et al. 2020) and cost models (Gregory and Lindsay 2016).

## Conclusion

Our work starts from the premise that the design of effective task-based interactive agents can be enhanced through incorporating observations of human interaction. For this purpose we are currently in the process of conducting our second user study to further investigate human response during interaction with a plan-based agent. In this work we have designed a layered framework aimed towards supporting task based human agent interaction, which abstracts the human agent dialogue aspect of the task from the decision task. Transaction dialogue/action models are used to capture the possible sequences for each of the agent's dialogue actions. We anticipate that much of what we learn in the user study can be directly incorporated into these transaction models and used to organise and extend them. Within the framework we have also considered how the agent's knowledge of the user can be developed during the planning task. In particular, we outline an approach we have developed for supporting within task elicitation, where the user's preferences can be discovered during execution and used to influence the executed action sequence.

## Acknowledgements

This work was funded by the UK's EPSRC Human-Like Computing programme under grant number EP/R031045/1, and by the ORCA Hub (`orcahub.org`), under EPSRC grant EP/R026173/1.

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
