# OpenReview forum: "Supporting Explanations Within an Instruction Giving Framework"
_icaps-conference.org/ICAPS/2021/Workshop/XAIP — XAIP 2021_

### Official Review · AnonReviewer2 · 2021-07-05
**Interesting and fleshed-out framework for human-agent interaction; I'm curious to know more about how to generalize it to new problems.**

**Rating:** 7
**Confidence:** 2

**Review:**

This paper presents a framework that allows interactions between an agent and a human user by creating abstractions that separate different components of a plan.  This allows the user to clarify or make modifications to a single component at a time.  This also allows the user to input preferences that can be taken into consideration to personalize the plan during execution.  The authors describe a fleshed out prototype system in a tourist guide domain that allows them to demonstrate their system.  The framework is inspired by analyzing an existing dataset of unman-human interactions in a similar domain that have been tagged and categorized by interaction type, suggesting that it is based on reasonable assumptions about how people might interact with agents.

The prototype presented in this paper seems like a relatively complete system that can realistically support this kind of complex human-agent interaction described in this domain. It's really interesting to see how all of the different components of this system fit together and allow for things like explanation and personalization, which are both really important.  I also appreciate that the interactions in this system are based on analyzing data from human-human interactions since this suggests that these are natural ways for humans to interact with each other, and plausibly also with agents by extension.

The main question I have about this paper is how well this framework will generalize beyond the specific domain it is described in here.  Will the assumption about the different steps being relatively independent hold in many other domains to allow for this kind of abstraction?  What happens if people communicate in ways that are different from what the agent expects?  How can the things that are hand-coded in this prototype be implemented (at least in theory)?  I'd find it compelling to have at least some initial hypotheses about some of these questions.

I also have a more minor writing comment/question: I was unsure exactly how important of a contribution the personalization piece is meant to be.  It sounded to me like a secondary contribution in the intro, but seemed like a major thing that the framework allows for (and a really interesting piece).

---

### Official Review · AnonReviewer1 · 2021-07-06
**A good, albeit somewhat preliminary, paper on a way of generating explanations for planning agents.**

**Rating:** 7
**Confidence:** 2

**Review:**

This paper describes a technique for generating explanations for tasks such as automated tour guides, inspired by related explanations from the HCRC Map Task. This includes considering the need to explain as a cost to favor explicability, and using information about the user's knowledge gained from previous interaction episodes to help estimate the explanation cost of plans.

In this reviewer's opinion, user studies (which are described in some detail as a topic of future work) would make this paper much stronger, although there is considerable discussion as to how explanations of the nature of those generated by the proposed system can be useful to the user of the system and this should suffice to warrant publication.

The paper is written pretty clearly and presents a nice survey of the relevant literature and the HCRC Map Task on which the work is based. This reviewer isn't sufficiently familiar with the literature to determine with confidence the originality or significance of this work.

---

### Meta-Review · Area_Chairs · 2021-07-08

**Recommendation:** Accept
**Confidence:** 5

**Metareview:**

The paper's a clear accept and a perfect match for the workshop.

---

### Decision · Program_Chairs · 2021-07-08

Accept